# Cannabis Use Moderates Methamphetamine- and HIV-Related Inflammation: Evidence from Human Plasma Markers

**DOI:** 10.3390/v17081143

**Published:** 2025-08-20

**Authors:** Jeffrey M. Rogers, Victoria O. Chentsova, Crystal X. Wang, Maria Cecilia Garibaldi Marcondes, Mariana Cherner, Ronald J. Ellis, Scott L. Letendre, Robert K. Heaton, Igor Grant, Jennifer E. Iudicello

**Affiliations:** 1San Diego State University/University of California San Diego Joint Doctoral Program in Clinical Psychology, San Diego, CA 92120, USA; vchentsova@health.ucsd.edu; 2Department of Psychiatry, University of California San Diego, San Diego, CA 92093, USAroellis@health.ucsd.edu (R.J.E.); jiudicello@health.ucsd.edu (J.E.I.); 3San Diego Biomedical Research Institute, San Diego, CA 92121, USA; cmarcondes@sdbri.org; 4Department of Neurosciences, University of California San Diego, San Diego, CA 92161, USA; 5Department of Medicine, University of California San Diego, San Diego, CA 92161, USA

**Keywords:** inflammatory signaling pathways, cannabinoids, methamphetamine, central nervous system stimulant, HIV, polysubstance use, substance use disorder

## Abstract

Background: Methamphetamine use, which is disproportionately prevalent among people with HIV, increases risk for cardio- and neurovascular pathology through persistent immune activation and inflammation. Preclinical studies indicate that cannabinoids may reduce markers of pro-inflammatory processes, but data from people with chronic inflammatory conditions are limited. We examined potentially interacting associations of lifetime methamphetamine use disorder (MUD), recent cannabis use, and HIV with four plasma markers of immune and inflammatory functions. Method: Participants with HIV (PWH, *n* = 86) and without HIV (PWoH, *n* = 148) provided urine and blood samples and completed neuromedical, psychiatric, and substance use assessments. Generalized linear models examined main and conditional associations of lifetime MUD, past-month cannabis use, and HIV with plasma concentrations of CXCL10/IP-10, CCL2/MCP-1, ICAM-1, and VCAM-1. Results: PWH displayed higher CXCL10/IP-10 than PWoH. Past-month cannabis use was independently associated with lower CXCL10/IP-10 levels and conditionally lower CCL2/MCP-1, ICAM-1, and VCAM-1 levels among people with lifetime MUD, but only PWoH displayed cannabis-associated lower VCAM-1 levels. Conclusions: Human plasma sample evidence suggests that cannabis use is associated with lower levels of immune and inflammatory molecules in the context of MUD or HIV. Cannabinoid pathways may be worthwhile clinical targets for treating sequelae of chronic inflammatory conditions.

## 1. Introduction

Methamphetamine (METH) is an illicit psychostimulant commonly used in the United States (US), and according to the National Survey on Drug Use and Health (NSDUH), US prevalence has more than doubled since 2015 (from 0.51% to 1.11%). METH poses a significant risk to public health, as research shows that METH is associated with cardiovascular disease [1] and cerebrovascular injury [2], among a host of other negative health outcomes [3,4,5]. Current evidence suggests that METH-related pathology is at least partially attributable to its negative influence on cerebrovascular endothelial cell [6,7,8] and peripheral immune functions [9,10]. METH is commonly used with other substances, and NSDUH estimates that 74.7% of past-year METH users also use cannabis [11], which has been studied for its potential to reduce the immune and inflammatory response in people with conditions characterized by chronic inflammation (e.g., HIV [12]). In our prior work [13,14], we found that in both people with HIV (PWH) and people without HIV (PWoH), recent cannabis use was associated with better neurocognitive functioning among people with METH use disorder (MUD).

While METH and cannabis have become increasingly prevalent in the US, use of these substances is known to be disproportionately prevalent among PWH, who may be especially impacted by METH-related pathology [15,16]. METH use, in particular, is more prevalent in PWH than in PWoH, as NSDUH 2023 estimates that PWH display 10.3% past-year METH use prevalence, compared with 0.8% in PWoH [17]. Importantly, the impact of METH use may differ in PWH, as both METH and HIV trigger pro-inflammatory mechanisms associated with the development of cardiovascular disease and peripheral immune dysfunction [12,18]. Cannabis is also used at substantially higher rates among PWH, estimated at 37.9%, compared with 19.0% among PWoH [19]. Before antiretroviral therapy (ART) advances, cannabis was commonly used to address symptoms associated with advanced HIV [20], but despite ART’s broad efficacy for improving health outcomes [21], quality of life [22], and overall survival [23] for PWH, chronic systemic inflammation remains a serious concern for comorbidity development [24,25,26]. Current evidence suggests that, despite modern advances in ART, treated PWH still display chronic and sustained activation of immune and endothelial cells [27,28,29,30].

Inflammatory and immune processes are commonly studied by measurable plasma markers of immune and endothelial activation. In an immune response, pro-inflammatory cytokines (e.g., interleukin-1/6 [IL-1, IL-6]) and chemokines (e.g., interferon-inducible protein 10 [CXCL10/IP-10]) are released from immune cells, initiating a cascade reaction among various molecules [31]. Adhesion molecules (e.g., ICAM-1, VCAM-1) interact with chemokines such as monocyte chemoattractant protein-1 (CCL2/MCP-1) to facilitate the recruitment, attachment, and transendothelial migration of circulating leukocytes [32,33,34], ultimately resulting in a chemokine-driven influx of activated monocytes into CNS tissue [35]. These mechanisms underlying chronic inflammatory processes are exacerbated by METH use, supported by animal models that have linked chronic METH exposure with increased levels of VCAM-1, ICAM-1, and CCL2/MCP-1 [36,37]. Similarly, a cross-species study demonstrated that METH-dependent mice and PWH recovering from METH dependence both displayed elevated levels of CCL2/MCP-1 and ICAM-1, compared to their counterparts with no history of METH exposure [38]. Studies have also linked increased levels of cytokines, chemokines, and cellular adhesion molecules (e.g., ICAM-1) to neurobehavioral disturbances in people with MUD [39,40].

Conversely, some cannabinoids (Δ9-Tetrahydrocannabinol [Δ9-THC] and cannabidiol [CBD]) have been shown to beneficially influence lymphocyte proliferation and inflammatory cytokine production. Pre-clinical models have shown that CB2 receptor activation reduces CD4 T-cell infection and HIV viral replication [41], and in humans, cannabis use may result in lower T-cell activation and lower levels of inflammatory monocytes and pro-inflammatory cytokines [42]. In people being treated for atherosclerosis, CBD and tetrahydrocannabivarin were shown to reduce VCAM-1 levels [43]. CBD has also been associated with VCAM-1 reductions and alleviation of psychomotor symptoms in people being treated for multiple sclerosis. In PWH, specifically, daily cannabis use has been linked to lower levels of immune activation and inflammation, as indicated by lower MCP-1 and CXCL10/IP-10 levels compared to PWH who do not use cannabis [44].

Current evidence suggests that repeated METH use and MUD increases risk for poor cardiovascular and neurocognitive outcomes by generating a chronic inflammatory state, and in some studies, cannabinoid administration has been shown to reduce markers of pro-inflammatory processes. Still, there is limited evidence regarding potential interactions between long-term/frequent METH use and cannabis effects on pro-inflammatory and immune functions in humans. It also remains unclear as to whether these substances may impact PWH and PWoH differently. The present study aims to examine the independent and potentially interacting influences of lifetime MUD, recent cannabis use (in the past month), and HIV on markers of inflammatory and immune functions. We hypothesized that (1) lifetime MUD and HIV are associated with higher levels of immune and inflammatory markers, and that (2) recent (past-month) cannabis use is more strongly associated with lower levels of immune and inflammatory markers among those with HIV and/or lifetime MUD relative to people without these conditions. Post hoc, we examined whether estimated total methamphetamine and cannabis exposure (e.g., lifetime amount and frequency of use) measures were associated with plasma marker levels within the METH- and cannabis-using subsamples.

## 2. Materials and Methods

We examined data from PWH on suppressive antiretroviral therapy and PWoH who had lifetime histories of METH use, cannabis use, both, or neither. Participants were recruited to National Institutes of Health-funded research studies conducted between 2009 and 2020 at the University of California San Diego HIV-Neurobehavioral Research Program. All participants provided written informed consent to procedures, which were approved by the UCSD Institutional Review Board. Participants were excluded from parent studies on the following criteria: history of American Psychiatric Association Diagnostic Statistical Manual IV (DSM-IV) psychotic disorders, and presence of any neurological conditions (e.g., stroke, epilepsy, history of moderate or severe traumatic brain injury). Exclusion criteria specific to this study included urine drug screen for substances other than METH or cannabis, DSM-IV alcohol dependence within one year of assessment, and other (non-cannabis, non-METH) substance abuse/dependence within one year of assessment. Lifetime DSM-IV alcohol abuse was permitted given the prevalence of use within this sample. PWH were excluded for HIV RNA load > 50 copies/mL in plasma or if they were not currently using antiretroviral (ARV) drug(s). The final analyzable sample consisted of 86 PWH on suppressive antiretroviral therapy and 148 PWoH.

### 2.1. Measures

#### 2.1.1. Neuromedical and Laboratory Assessments

Data were collected using standardized neuromedical, laboratory, neurocognitive, and psychiatric evaluations. Current and past medical conditions were collected via a comprehensive neuromedical interview using ICD-9 diagnostic codes. Complete blood counts, rapid plasma reagin, and CD4+ T-cells were measured using routine clinical chemistry panels. Rapid blood spot tests assessed hepatitis B and C infection. Breathalyzer assessment was used to screen recent alcohol use, and blood and urine specimens were used for routine clinical labs, diagnostic tests, and urine drug screening. No participants had a positive breathalyzer test on the morning of the evaluation.

Plasma biospecimens were collected using EDTA vacuum tubes and standard phlebotomy procedures. Soluble levels of immune and endothelial activation (i.e., chemokines CXCL10/IP-10 and CCL2/MCP-1, cellular adhesion molecules ICAM-1 and VCAM-1) were measured in duplicate on MesoScale Discovery Imager 2400 (Rockville, MD, USA), which uses traditional sandwich-based immunoassay concepts and electrochemiluminescence detection. A carbon surface plate was coated with a capture antibody against the target protein of interest. Samples were added to the plate with an electrochemiluminescent detection antibody, and electrochemical stimulation (620 nm) was then applied to the plate electrodes, causing the bound label to emit light, which was used to measure and quantify the target protein concentration. Assays were repeated on samples when the coefficient of variation was greater than 20%. Additionally, 10% of assays were selected at random for repetition to assess operator and batch consistency. To minimize batch effects, raw plasma marker data were standardized within analysis plates by scaling values to each plate’s median and median absolute deviation.

#### 2.1.2. HIV Disease and Treatment Characteristics

HIV serostatus was determined via self-report and confirmed with a finger stick point-of-care test (MedMira Inc., Halifax, NS, Canada). ARV usage history was measured using a structured, clinician-administered questionnaire. Levels of HIV viral load in plasma were measured using reverse transcriptase polymerase chain reaction (Amplicor, Roche Diagnostics, Indianapolis, IN, USA).

#### 2.1.3. Psychiatric and Substance Use History

The Composite International Diagnostic Interview [45] was administered to assess for current and lifetime substance use and mood disorders (e.g., major depressive disorder) based on DSM-IV criteria. Detailed lifetime cannabis and METH use characteristics, including age at first use, years since last use, age at first cannabis/METH use disorder, years since last use disorder, and estimated total lifetime duration of use (days) and total lifetime quantity (grams) were gathered using a semi-structured timeline, follow-back substance use interview [46].

### 2.2. Data Analysis

All analyses were conducted using R (version 4.3.2; [47]). We aimed to examine the degree to which HIV status, METH use, and cannabis use are associated with plasma markers of immune and endothelial activation. All statistical models described below are multivariable linear regression models, estimated using heteroscedasticity-robust standard errors [48]. Model parameter estimates provided are standardized beta coefficients (B; mean-centered, standard deviation-scaled) and model variance explained (*R*^2^). Response variables were within-plate, median absolute deviance-scaled plasma markers of immune and endothelial activation (VCAM-1, ICAM-1, CXCL10/IP10, and CCL2/MCP-1). Of the included participants, four (1.7%) were missing plasma marker data, which were assumed to be missing at random for modeling purposes.

Primary explanatory variables of interest were reference group contrasts for HIV status, lifetime MUD, and past-month cannabis use. To determine covariate inclusion in our base (covariates-only) model, we initially examined demographic, health factors, and other substance use variables’ (see Table 1) association with VCAM-1, ICAM-1, CXCL10/IP10, and CCL2/MCP-1. We retained all covariates that were significantly associated with at least one plasma marker; all models initially included age, sex, ethnicity, education years, lifetime alcohol use disorder, and past-month tobacco use. Main associations of HIV status (PWoH/PWH), lifetime MUD (MUD−/MUD+), and past-month cannabis use (C−/C+) were added to the best fitting covariates-only model. We then examined whether the associations of lifetime MUD and past-month cannabis use were conditional upon one another and/or were conditional on HIV status. Significant conditional effects were modeled using marginal comparisons to the negative–negative group (i.e., MUD−C− vs. MUD+C−, MUD−C− vs. MUD−C+, MUD−C− vs. MUD+C+).

We further examined possible relationships between METH and cannabis use characteristics and variations in plasma marker levels within subsamples who used either METH (*n* = 123, 40.7% PWH) or cannabis (*n* = 168, 38.1% PWH) at least five times in their life. Substance exposure variables, estimated from timeline follow-back interview, for METH and cannabis were lifetime frequency of use (total days, log_10_-transformed), total amount used (grams, log_10_-transformed), recency of use (days since last used), and age at first use. With the best fitting base models from the prior modeling stage, we added substance use exposure variables independently, as indices of substance exposures displayed significant covariance, indicated by model variance inflation factors. We additionally tested for conditional effects of HIV status on associations between substance use exposure variables and plasma markers.

## 3. Results

### 3.1. Participant Characteristics

Sample descriptive characteristics are provided in Table 1 for subsamples of PWH (*n* = 86), PWoH (*n* = 148), and for the overall sample. Overall, participants were 42.1 ± 13.6 years of age on average, and the subsample of PWH was older than PWoH (44.7 vs. 40.5 years, *p* = .023). A majority of participants were male (73.5%), and the sample of PWH was more disproportionately male (93.0% vs. 62.2%, *p* < .001). The racial/ethnic composition of the sample was 52.6% White, 24.4% Hispanic, 15.8% Black, and 7.3% other, and these proportions were not significantly different between PWH and PWoH (*p* = .829). Participants reported an average of 13.8 ± 2.5 years of education, and PWH reported significantly more years of education (14.3 vs. 13.6, *p* = .030). A minority of participants met clinical criteria for hypertension (17.5%), hyperlipidemia (15.8%), and diabetes (3.8%); hyperlipidemia was more common in PWH (24.4% vs. 10.8%, *p* = .006). Lifetime major depressive disorder criteria were met by 32.5% and were more common in PWH (41.9% vs. 27.0%, *p* = .019). Past-month tobacco use was reported by 42.7%, and rates did not differ between PWH and PWoH (*p* = .451). The most common lifetime DSM-IV substance use disorder (abuse/dependence) diagnoses were alcohol (47.0%), METH (45.7%), cannabis (30.8%), and cocaine (14.5%), and rates of these did not differ between PWH and PWoH.

#### 3.1.1. HIV Treatment History and Indices of Severity

Descriptive data specific to the subsample of PWH are displayed in Table 2. All participants with HIV (*n* = 86) were currently on antiretroviral therapy and virally suppressed (i.e., HIV plasma RNA values less than or equal to 50 copies/mL). Mean plasma RNA levels were 11.8 ± 17.4 (median = 0.0, IQR = 33.0). The average duration of participants’ current ARV regimen was 29.3 ± 38.7 months (median = 13.5, IQR = 32.6). Average lifetime Nadir CD4^+^ T-cell count was 282.0 ± 195.1 (range = 3–800), and 44.2% (*n* = 38/86) had been diagnosed with acquired immunodeficiency syndrome (AIDS) in their lifetime.

#### 3.1.2. Cannabis Use

Cannabis use descriptive data are displayed in Table 3 for *n* = 168 (71.8%) participants with >5 lifetime cannabis use events. Participants reported a median ± IQR of 961.5 ± 2750.5 days of use, 340.3 ± 3131.0 total grams used, 334.8 ± 2476.6 days since last cannabis use, and that their average age at first use was 16.0 ± 4.0 years of age. Past-month cannabis use was reported by 32.7%, and lifetime and past-month cannabis use disorder rates were 42.9% and 3.0%, respectively. None of these factors were significantly different between PWH and PWoH (*p*s = .171–.473). Participants who used cannabis in the past month displayed greater lifetime days of use (3324.0 vs. 1842.8 days, *p* = .007), greater lifetime quantity of use (4220.179 vs. 2189.6 g, *p* = .020), and younger age at first use (15.1 vs. 17.3, *p* = .012) than participants who did not use cannabis in the past month.

#### 3.1.3. Methamphetamine Use

METH use descriptive data are displayed in the bottom section of Table 3 for participants reporting METH use >5 times in their life (*n* = 123, 52.6%). Participants reported a median ± IQR of 1563.0 ± 2611.5 days of use, 1193.0 ± 2689.0 total grams used, and 61.0 ± 168.6 days since last METH use. Participants first used METH at 23.0 ± 14.5 years of age. Thirty-five percent reported using METH in the past month, and 87.0% met criteria for lifetime MUD. PWoH displayed greater lifetime quantity of use than PWH (median = 1330.5 vs. 1065.9, *p* = .049), but no other lifetime METH characteristics differed between PWH and PWoH (*p*s = .110–.678).

### 3.2. Modeled Associations with Plasma Biomarkers

Results for models of soluble CXCL10/IP-10, CCL2/MCP-1, ICAM-1, and VCAM-1 are displayed in Table 4. Covariates retained in each model displayed the following statistically significant associations: Older age was independently associated with higher levels of ICAM-1 (B = 0.11, *p* = .016), IP-10 (B = 0.11, *p* = .014), MCP-1 (B = 0.31, *p* < .001). Male sex was associated with greater MCP-1 levels (B = 0.40, *p* = .014). Greater education years were associated with lower levels of ICAM-1 (B = −0.12, *p* = .004). Compared with White participants, US minority participants displayed lower VCAM-1 levels (B = −0.35, *p* < .001). Past-month tobacco use was associated with greater levels of ICAM-1 (B = 0.26, *p* = .014). Lifetime DSM-IV alcohol use disorder and lifetime major depressive disorder was not significantly associated with examined plasma marker levels.

#### 3.2.1. CXCL10/IP-10

CXCL10/IP-10 levels displayed significant main associations with HIV status and past-month cannabis use, with HIV status being associated with higher CXCL10/IP-10 levels (B = 0.54, *p* < .001) and past-month cannabis use being associated with lower CXCL10/IP-10 levels (B = −0.33, *p* = .004). Lifetime MUD was not significantly associated with CXCL10/IP-10 levels. Models examining interactions between HIV status, lifetime MUD, and/or past-month cannabis use found no significant conditional associations. Main associations of HIV status and past-month cannabis use with CXCL10/IP-10 levels are displayed in Figure 1.

#### 3.2.2. CCL2/MCP-1

CCL2/MCP-1 levels displayed a significant main association with lifetime MUD, such that people with MUD displayed higher levels of CCL2/MCP-1 (B = 0.32, *p* = .023). HIV status and past-month cannabis use displayed no significant main associations with CCL2/MCP-1. Conditional effects models found that the association between MCP-1 and lifetime MUD was conditional on past-month cannabis use (B = −0.23, *p* = .037). Simple comparisons to participants without lifetime MUD or past-month cannabis use (MUD−C−) found that those with MUD but not recent cannabis use (MUD+C−) displayed significantly greater CCL2/MCP-1 levels (B = 0.36, *p* = .030), but participants with both (MUD+C+) displayed relatively lower and non-significant differences in CCL2/MCP-1 (B = 0.21, *p* = .257). Associations of lifetime MUD and past-month cannabis use with CCL2/MCP-1 were not conditional on HIV status. Model-fitted CCL2/MCP-1 distributions stratified by MUD and past-month cannabis use are displayed in the top panel of Figure 2.

#### 3.2.3. ICAM-1

ICAM-1 displayed no significant main associations with lifetime HIV status, MUD, or past-month cannabis use. Conditional effects models found a significant interaction between lifetime MUD and past-month cannabis use (B = −0.27, *p* = .013), independent of HIV status. Simple comparisons to participants without lifetime MUD or past-month cannabis use (MUD−C−) found that those with MUD but not recent cannabis use (MUD+C−) displayed significantly greater ICAM-1 levels (B = 0.26, *p* = .007), but participants with both (MUD+C+) displayed relatively lower and non-significant differences in ICAM-1 (B = −0.00, *p* = .987). Model-fitted ICAM-1 distributions stratified by MUD and past-month cannabis use are displayed in the bottom panel of Figure 2.

#### 3.2.4. VCAM-1

VCAM-1 displayed significant main associations with HIV status and lifetime MUD, such that HIV status (B = 0.31, *p* < .001) and lifetime MUD (B = 0.25, *p* = .002) were independently associated with higher levels of VCAM-1. VCAM-1 displayed no significant main associations with past-month cannabis use. Conditional effects models found a significant three-way interaction between lifetime HIV status, MUD, and past-month cannabis use. Simple comparisons to participants without lifetime MUD or past-month cannabis use (MUD−C−), modeled as being conditional on HIV status, found that those with past-month cannabis use without MUD (MUD−C+) displayed significantly lower VCAM-1 than MUD−C− in PWH (B = −0.61, *p* = .012) but not in PWoH (B = 0.19, *p* = .258). People with MUD but not recent cannabis use (MUD+C−) displayed significantly greater VCAM-1 than MUD−C− in PWoH (B = 0.42, *p* < .001), but not in PWH (B = −0.44, *p* = .010). Participants with both MUD and past-month cannabis use (MUD+C+) displayed not-significant differences with MUD−C− in both PWH and PWoH. Figure 3 displays boxplot VCAM-1 distributions, split by MUD and recent cannabis use, and plotted separately for PWH (panel 1) and PWoH (panel 2).

### 3.3. Associations Between Lifetime Cannabis and METH Use Characteristics and Plasma Biomarkers

Age at first cannabis use was significantly associated with lower CXCL10/IP-10 levels (B = 0.11, *p* = .024), such that lower (younger) ages at first use were associated with lower IP-10 levels. Age at first use was significantly different among people with and without past-month cannabis use, and after adding past-month cannabis use to the model, age at first use was no longer significantly associated with CXCL-10/IP-10. Total lifetime days of cannabis use and total lifetime grams of cannabis used were not significantly associated with any examined plasma biomarkers.

Lifetime days of METH use and total grams used displayed significant main associations with plasma CCL2/MCP-1 levels, such that greater days of use (B = 0.17, *p* = .029) and greater lifetime grams used (B = 0.23, *p* = .014) were associated with greater CCL2/MCP-1 levels. These associations were found to be conditional on past-month cannabis use, such that only participants with past-month cannabis use displayed greater CCL2/MCP-1 at the highest levels of lifetime METH use and lower CCL2/MCP-1 at lower levels of lifetime METH use. These conditional associations with CCL2/MCP1 are displayed in Figure 4. Days since last METH use and age at first METH use were not significantly associated with any of the examined plasma biomarkers.

## 4. Discussion

To better understand the biological processes by which METH and HIV disease confer risk for adverse cardiovascular, cerebrovascular, and cognitive outcomes, and whether cannabis might exert an ameliorative influence, we analyzed plasma markers linked to vascular disorders underlying inflammation such as CXL10/IP-10, CCL2/MCP-1, VCAM-1, and ICAM-1 in people with and without METH use disorder (MUD), past-month cannabis use, and HIV. Preclinical and clinical studies have previously linked MUD and HIV disease to inflammation and associated processes (e.g., vascular dysfunction), while most pre-clinical evidence has shown that cannabis may reduce inflammation in people with pro-inflammatory conditions [43,49,50]. Supported by these studies and our prior work examining neurocognitive outcomes in people who use METH and cannabis, we hypothesized that (1) lifetime MUD and HIV status will be associated with inflammation and endothelial activation, as measured by higher plasma levels of CXL10/IP-10, CCL2/MCP-1, VCAM-1, and ICAM-1, and (2) that recent (i.e., past-month) cannabis use would be associated with reduced plasma marker levels across groups.

Our results support the hypothesis that HIV and MUD are associated with increased immune activation, systemic inflammation, and endothelial activation, as we found elevated markers of CCL2/MCP1, VCAM-1, and ICAM-1 among those with lifetime MUD and elevated markers of CXCL10/IP-10 and VCAM-1 among PWH. While MUD and HIV have been linked to increased levels of these markers in prior studies, models of CCL2/MCP-1 and ICAM-1 supported the more novel hypothesis that cannabis use may be associated with lower levels of these markers in people with MUD, as we found conditionally lower levels of CCL2/MCP-1 and ICAM-1 in those with lifetime MUD and past-month cannabis use. Conversely, participants with lifetime MUD but who were not currently using cannabis displayed elevated CCL2/MCP-1 and ICAM-1. Further, HIV status differentiated the influence of cannabis use for ICAM-1, as only PWoH displayed lower plasma levels in the setting of recent cannabis use and lifetime MUD. Our model of CXL10/IP-10 levels also broadly supported our second hypothesis, as recent cannabis use was independently associated with lower plasma marker levels.

### 4.1. CXCL10/IP-10

IP-10 is a chemokine that attracts activated T-cells to the sites of inflammation and is commonly elevated in conditions characterized by chronic inflammation (e.g., HIV). Among PWH, and even those on ARVs and who were virally suppressed, elevated IP-10 levels are strongly associated with adverse outcomes in PWH, including neurocognitive impairment [51,52]. Consistent with prior work, we found that PWH on suppressive ARVs displayed higher CXCL10/IP-10 than PWoH, independent of MUD and recent cannabis use.

Additionally, we found that past-month cannabis use was associated with lower IP-10 levels, regardless of MUD or HIV status. These data align with prior studies linking recent cannabis use to lower plasma IP-10 levels in PWH [42,53,54] and support the notion that cannabinoids may modulate inflammatory processes in PWH. The current study, to our knowledge, is among the first to show this association with PWoH. Of note, we did not find an association between MUD and higher IP-10, which has been observed in other studies [18], possibly due to the large influence of HIV status in those models.

### 4.2. CCL2/MCP-1

MCP-1 is a chemokine that attracts immune cells to sites of infection, injury, or inflammation, and has been implicated in the pathogenesis of HIV and other diseases (e.g., atherosclerosis [55]). Additionally, existing evidence shows that MCP-1 is elevated in response to both chronic and acute METH use, via immune cell activation and central nervous system astrocytes, which release MCP-1 into circulation [36,37,38]. METH-associated increases in MCP-1 are posited to increase monocyte trafficking into the CNS, contributing to adverse downstream effects (e.g., neuroinflammation, neurocognitive impairment [2,34]). In this study, we found that MCP-1 was elevated among people with MUD, regardless of HIV status, but only in those who did not recently use cannabis. MCP-1 levels in recent cannabis users alone and in the context of MUD, were like those of the comparison group (M-C-).

Our findings provide novel support to the hypothesis that cannabis may exert an immunomodulatory effect, possibly via CB2 activation [41,54], reducing monocyte recruitment, and immune activation. Of note, we did not find an MCP-1 elevation related to HIV status as has been observed in other studies [56,57,58], possibly due to the fact that PWH in our study were on ART and virally suppressed, as MCP-1 levels normalize more readily with ART and without ongoing viral replication and inflammatory triggers [59]. Further context from METH subsample analyses suggests that recent cannabis use was associated with lower MCP-1 for those with relatively lower amounts of lifetime METH use (~1100 total days or less). Cannabis-associated lower MCP-1 levels may be specific to moderate or lower levels of MUD.

### 4.3. VCAM-1

VCAM-1 is an adhesion molecule expressed on endothelial cells in response to inflammatory cytokines and facilitates the adhesion and migration of leukocytes across the endothelium [60]. Higher levels of plasma VCAM-1 have been linked to HIV [61] and METH [36,37], as well as inflammation, vascular dysfunction, blood–brain barrier dysfunction, and neurocognitive impairment [61,62,63]. Consistently, research has found HIV disease processes are associated with VCAM-1 upregulation [64,65,66], and preclinical/translational models have found acute METH exposure induces oxidative stress responses, thereby upregulating VCAM-1 expression. Early evidence from pre-clinical studies have identified associations between CB2 receptor agonism and reduced VCAM expression [67,68,69].

In the present study VCAM-1 model results displayed the greatest complexity, as associations were dependent upon MUD, recent cannabis use, and HIV status. First, our results are largely consistent with the hypotheses that METH and HIV disease increase VCAM-1 levels; however, we found that recent cannabis use was only associated with lower levels of VCAM-1 in the presence of only one of these two conditions. In the setting of both MUD and HIV, recent cannabis use was not associated with lower VCAM-1. Furthermore, in PWH, recent cannabis use was associated with reduced plasma VCAM-1 levels, but only in those without MUD, suggesting that recent cannabis use may mitigate endothelial activation in this subset of PWH.

Our results provide novel insights from human plasma data that cannabis is associated with lower VCAM-1 levels, but only in the context of MUD or HIV, not both. One possibility is that, in the context of both HIV and METH dependence, VCAM-1 expression may be driven through multiple pathways (e.g., HIV-associated immune activation versus METH-associated oxidative stress), thereby decreasing the potential for cannabis to be associated with lower VCAM-1. Alternatively, the effects of cannabis on one or both pathways may be dependent on cannabis use characteristics (e.g., dose, type of cannabinoids used), as prior work has focused specifically on cannabidiol (CBD) and CB2 receptor agonism [50,67,68,69]. We did not collect data on the specific cannabis products used by participants and were unable to test hypotheses related to CBD exposure. Because our participants were recreational cannabis users in the southwestern US, who largely began using cannabis prior to the availability of legal dispensaries, Δ9-tetrahydrocannabinol (Δ9-THC) is likely the primary cannabinoid they consumed, followed distantly by CBD and tetrahydrocannabivarin [70,71]. There is notable evidence of Δ9-THC displaying greater affinity for both CB1 and CB2 receptors than the other most common phytocannabinoids, implicating a wider array of potential cannabinoid pathways than previous research on CB2 receptor agonism, specifically.

Lastly, in PWoH, recent cannabis use was associated with lower VCAM-1 in the context of MUD, providing novel human plasma evidence to support the hypothesis that METH-driven endothelial activation may be moderated by cannabis use in PWoH.

### 4.4. ICAM-1

ICAM-1, like VCAM-1, is an adhesion molecule expressed following exposure to pro-inflammatory cytokines. VCAM-1 expression is more specific to cardiovascular endothelium and is more directly linked to chronic immune activation, whereas ICAM-1 is expressed more broadly in various cell types (e.g., endothelial and epithelial cells, macrophages, etc.).

We found a similar pattern of results for ICAM-1 as with MCP-1—that people with MUD who did not recently use cannabis displayed elevated ICAM-1 levels, whereas those who recently used cannabis displayed comparable ICAM-1 levels to the comparison group (M-C-). We did not find significant associations between HIV status and ICAM-1. Divergence between VCAM-1 and ICAM-1, especially regarding the influence (or lack thereof) of HIV status, may be at least partially attributable to ICAM-1’s relatively wider signaling array. In addition, unlike VCAM-1, which can remain elevated despite ART and viral suppression, ICAM-1 levels have been shown to be reduced with effective ART [72]. Our finding that ICAM-1 was elevated with METH use in the absence of recent cannabis use supports prior findings that METH exacerbates cardiovascular risk via immune cell activation and the production of reactive oxygen species, and that cannabis use may ameliorate aspects of these pro-inflammatory processes. Of note, past-month tobacco use was also strongly associated with higher levels of ICAM-1 in our sample, highlighting a modifiable target for intervention as cessation is associated with markedly reduced ICAM-1 concentrations [73].

### 4.5. Limitations and Future Directions

Our sample size was sufficient for detecting moderate effect sizes in simple cross-over interactions using a single set of a priori-specified contrasts, but our ability to detect moderate effects in multi-level (3-way) interactions was negatively impacted by a limited sample of PWH who have MUD and who used cannabis in the past month. This concern was partially ameliorated with good model fit indices for most models, resulting in more precise estimates and greater power, but there may be legitimate associations that were not detected in our analysis. In terms of generalizability, our ability to detect and draw inferences from estimates related to biological sex among PWH was limited by sample size (7% female).

Regarding the operationalization and examination of substance use variables, we did not have information about the cannabinoid products our participants were using at the time of their reports. Cannabinoid composition (e.g., Δ9-THC vs. CBD vs. other cannabinoid dominance) may influence the degree to which cannabis is associated with changes in plasma markers (e.g., Δ9-THC extract products and whole plant matter produce unique constellations of biological responses). Furthermore, conclusions herein may be specific to our analytic strategy of modeling recent cannabis use (i.e., in past month). Our sample contained few people with current cannabis use disorder, and it is plausible that a current use disorder may be associated with different plasma marker relationships than current non-use disorder levels of cannabis use. This concern is partially ameliorated by the fact that our past-month cannabis use subsample displayed greater lifetime amount and days of cannabis use and earlier ages at first cannabis use than those who did not use cannabis in the past month.

Importantly, there is currently a wide variation in commonly used cannabis products, and it will be important for future research to elucidate the pharmacological differences between products and whether this influences subsequent biological cascades. Participants in this sample were all using cannabis recreationally prior to the availability of legal dispensaries, and data from such cannabis samples indicates that Δ9-THC was the predominant cannabinoid, with CBD levels being very low or undetectable [70,71].

The evidence surrounding substance use influences on pro-inflammatory processes would be improved from analysis of paired plasma and cerebrospinal fluid markers, as results would more directly reflect central nervous system outcomes. Paired samples may also provide the potential for unique profiles of immune and inflammatory markers to emerge. Lastly, given the cross-sectional nature of this study, we are unable to establish causality. Future longitudinal and/or mechanistic studies may provide additional insights into these complex findings.

## 5. Conclusions

METH use disorder is highly prevalent in PWH, and both can have significant effects on immune function and pro-inflammatory processes that lead to significant central nervous system consequences, despite modern advances in anti-retroviral therapy effectiveness and tolerability. Results from this study support prior findings that METH and HIV disease confer risk for negative outcomes via their influence on chronic inflammatory processes, and we provide novel evidence from human plasma samples that cannabis use is associated with reduced levels of immune and inflammatory molecules in the context of chronic METH use or HIV infection (CCL2/MCP-1, VCAM-1, ICAM-1) and independent of METH use and HIV (CXCL10/IP-10). Associations between cannabis use and lower indices of inflammatory pathology from HIV and MUD point toward cannabinoid pathways as promising therapeutic targets that warrant further study.

## Figures and Tables

**Figure 1 viruses-17-01143-f001:**
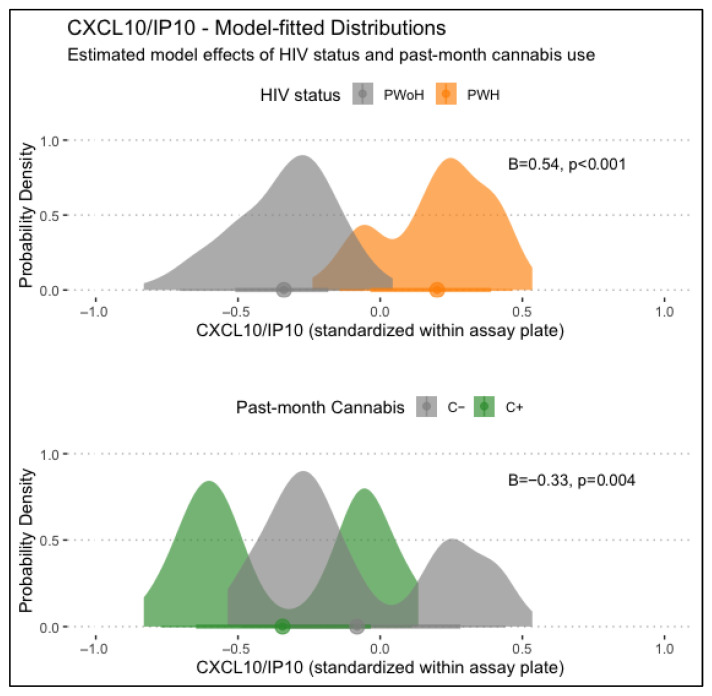
Displayed are main associations of HIV status and past-month cannabis use with CXCL10/IP-10 levels, estimated from generalized linear regression models. PWH displayed significantly higher CXCL10/IP10 levels than PWoH (B = 0.54, *p* < .001), and those with past-month cannabis use displayed significantly lower levels (B = −0.33, *p* = .004).

**Figure 2 viruses-17-01143-f002:**
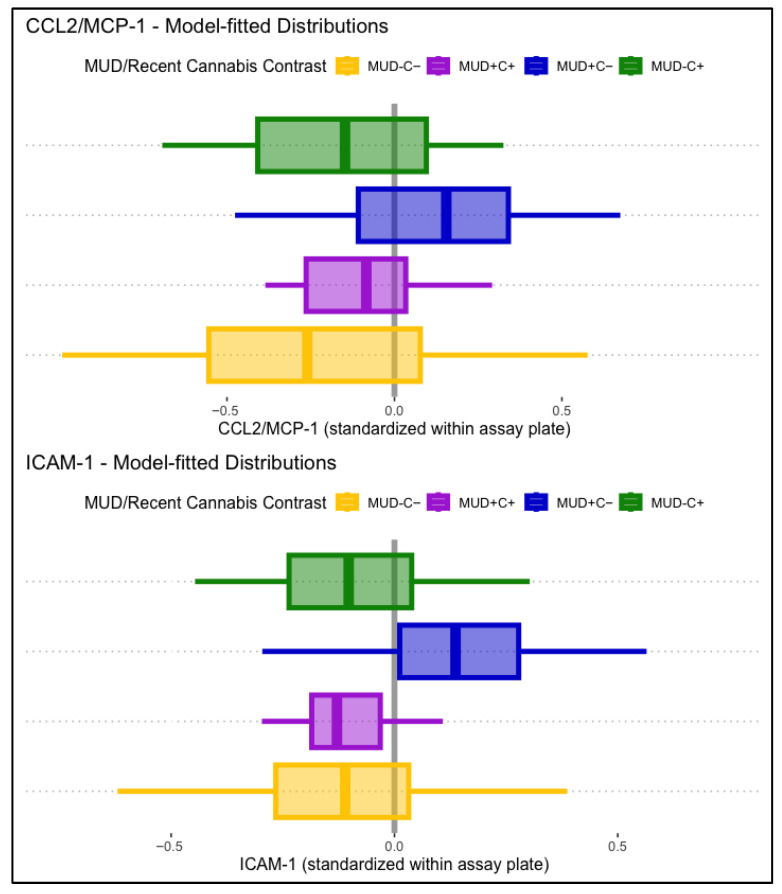
Displayed are model-fitted distributions of CCL2/MCP-1 (**top** panel) and ICAM-1 (**bottom** panel), split by lifetime MUD (+/−) and recent cannabis use (C+/C−). Participants with MUD but not recent cannabis use (MUD+C−, blue) displayed significantly higher CCL2/MCP-1 (B = 0.36, *p* = .030) and ICAM-1 (B = 0.26, *p* = .007) levels than the comparison group (MUD−C−, yellow).

**Figure 3 viruses-17-01143-f003:**
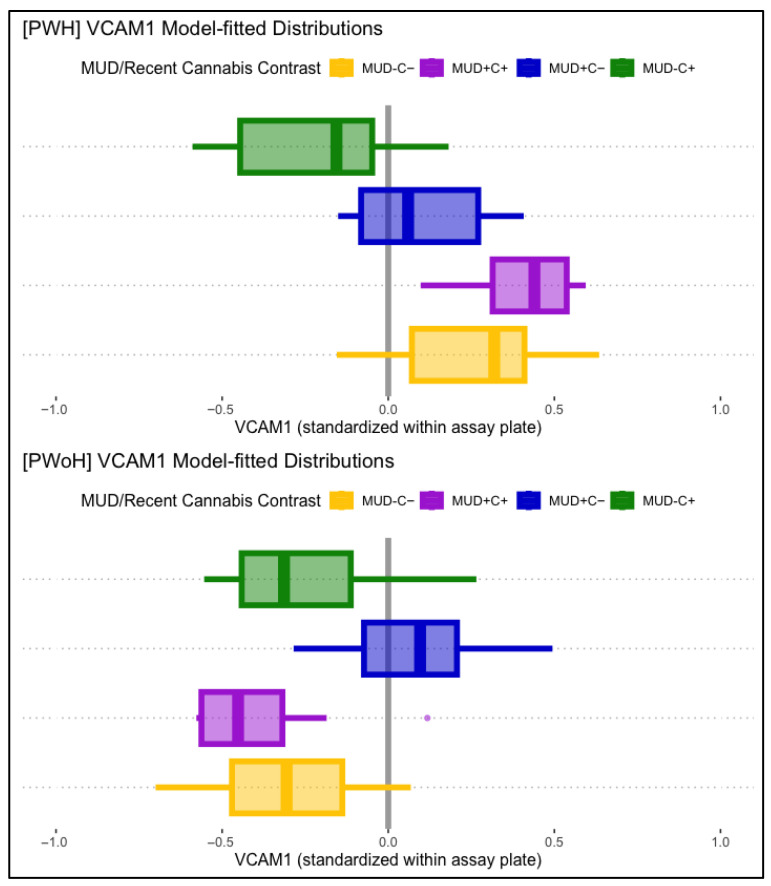
Displayed are model-fitted VCAM-1 distributions, split by lifetime MUD (+/−) and recent cannabis use (C+/C−), as well as HIV status (PWH in top panel, PWoH in bottom panel). Those with past-month cannabis use (MUD−C+, green) displayed lower VCAM-1 than the comparison group (MUD−C−, yellow) in PWH (**top** panel), but not in PWoH (**bottom** panel). Those with MUD but not recent cannabis use (MUD+C−, blue) displayed significantly higher VCAM-1 levels in PWoH (**bottom** panel), but not in PWH (**top** panel).

**Figure 4 viruses-17-01143-f004:**
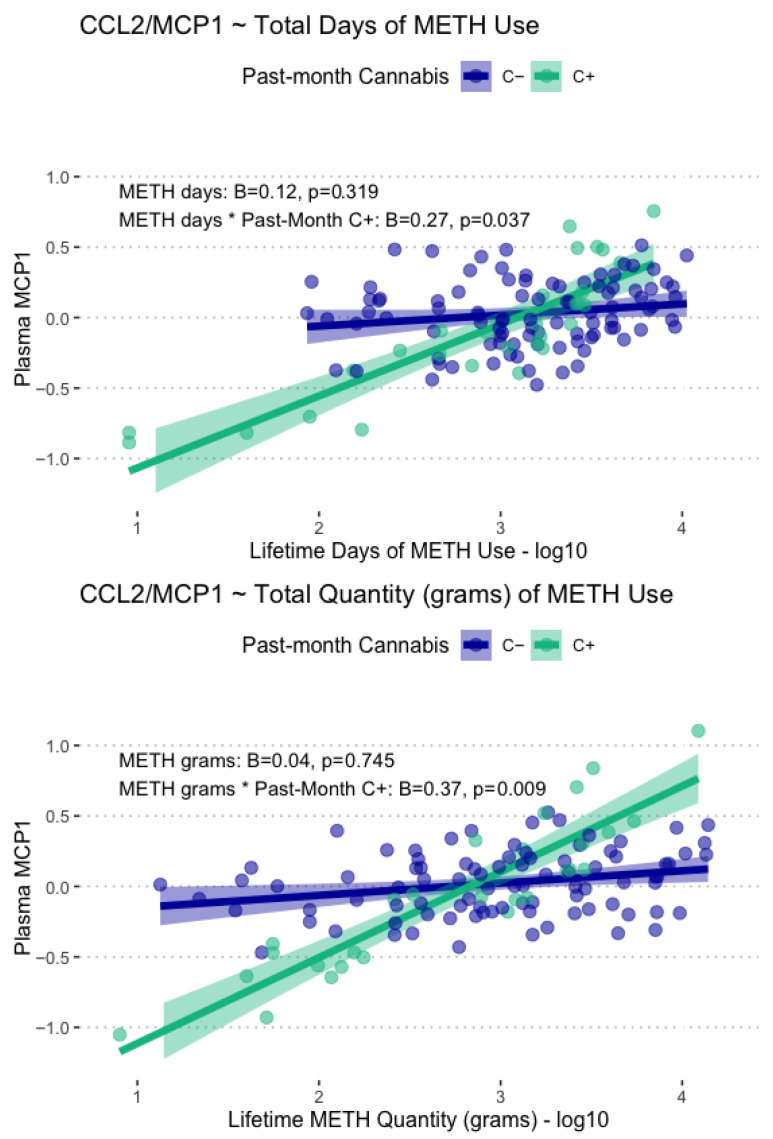
Displayed are model-fitted conditional associations between total lifetime days and grams of METH use, past-month cannabis use, and plasma levels of CCL2/MCP-1. Total METH days and total METH grams were only associated with CCL2/MCP1 in those with past-month cannabis use (C+, green).

**Table 1 viruses-17-01143-t001:** Sample demographic and descriptive data for people with (PWH) and without (PWoH) HIV and for the total sample. Reported *p*-values (*p*) were determined from chi-square difference tests for categorical variables and heteroscedasticity robust *t*-tests for continuous variables.

	PWoH (*n* = 148)	PWH (*n* = 86)	Total (*n* = 234)	*p*
Age				.023
Mean	40.53	44.70	42.06	
SD	14.15	12.20	13.59	
Sex				<.001
Female	56 (37.8%)	6 (7.0%)	62 (26.5%)	
Male	92 (62.2%)	80 (93.0%)	172 (73.5%)	
Ethnicity				.829
White	76 (51.4%)	47 (54.7%)	123 (52.6%)	
Hispanic	35 (23.6%)	22 (25.6%)	57 (24.4%)	
Black	25 (16.9%)	12 (14.0%)	37 (15.8%)	
Other	12 (8.1%)	5 (5.8%)	17 (7.3%)	
Education Years				.03
Mean	13.57	14.30	13.84	
SD	2.48	2.42	2.48	
Hypertension	21 (14.2%)	20 (23.3%)	41 (17.5%)	.079
Hyperlipidemia	16 (10.8%)	21 (24.4%)	37 (15.8%)	.006
Diabetes	6 (4.1%)	3 (3.5%)	9 (3.8%)	.828
Lifetime Major Depressive Disorder	40 (27.0%)	36 (41.9%)	76 (32.5%)	.019
Past-month Tobacco Use	66 (44.6%)	34 (39.5%)	100 (42.7%)	.451
Past-month Cannabis Use	30 (20.3%)	25 (29.1%)	55 (23.5%)	.126
Past-month METH Use	27 (18.2%)	16 (18.6%)	43 (18.4%)	.945
Lifetime DSM-IV Cannabis	47 (31.8%)	25 (29.1%)	72 (30.8%)	.668
Lifetime DSM-IV METH	65 (43.9%)	42 (48.8%)	107 (45.7%)	.467
Lifetime DSM-IV Alcohol	76 (51.4%)	34 (39.5%)	110 (47.0%)	.081
Lifetime DSM-IV Cocaine	20 (13.5%)	14 (16.3%)	34 (14.5%)	.563
Lifetime DSM-IV Hallucinogen	11 (7.4%)	7 (8.1%)	18 (7.7%)	.845
Lifetime DSM-IV Opioid	14 (9.5%)	5 (5.8%)	19 (8.1%)	.325
Lifetime DSM-IV Sedative	8 (5.4%)	6 (7.0%)	14 (6.0%)	.625

SD = standard deviation of the mean, DSM-IV = American Psychiatric Association Diagnostic Statistical Manual IV.

**Table 2 viruses-17-01143-t002:** HIV disease and treatment characteristics for the subsample of PWH (*n* = 86).

PWH (*n* = 86) Sample Characteristics	
Months on Current ARV Regimen	
Mean (SD)	29.3 (38.7)
Median (IQR)	13.5 (32.6)
HIV Plasma RNA (copies/mL)	
Mean (SD)	11.8 (17.4)
Median (IQR)	0.0 (33.0)
Current CD4^+^ T cell count	
Mean (SD)	668.6 (257.9)
Median (IQR)	642.0 (342.0)
Nadir CD4^+^ T cell count	
Mean (SD)	282 (195.1)
Median (IQR)	275 (271.8)
AIDS Diagnosis	
* n* (%)	38 (44.2%)

ARV = antiretroviral (drug), SD = standard deviation of the mean, IQR = interquartile ratio, AIDS = acquired immunodeficiency syndrome.

**Table 3 viruses-17-01143-t003:** Substance characteristics for the subset of PWoH and PWH who used cannabis > 5 times in their life (*n* = 168 total), and people who used METH > 5 times in their life (*n* = 123 total). Reported *p*-values (*p*) reported are from chi-square difference tests for categorical variables and heteroscedasticity robust *t*-tests for continuous variables.

**Cannabis Use Characteristics (*n* = 168)**	**PWoH (*n* = 104)**	**PWH (*n* = 64)**	**Total (*n* = 168)**	** *p* **
Past-month Cannabis Use	30 (28.8%)	25 (39.1%)	55 (32.7%)	0.171
Lifetime DSM-IV Cannabis Use Disorder	47 (45.2%)	25 (39.1%)	72 (42.9%)	0.436
Lifetime Days of Cannabis Use				0.295
Mean (SD)	2115.7 (3206.6)	2672.3 (3535.0)	2327.7 (3336.1)	
Median (IQR)	750.5 (2534.5)	1188.0 (4007.0)	961.5 (2750.5)	
Lifetime Cannabis Quantity (grams)				0.473
Mean (SD)	3238 (6767.7)	2552.1 (4457.7)	2976.7 (5988.1)	
Median (IQR)	230.7 (2657.2)	412.6 (3456.6)	340.3 (3131)	
Days Since Last Cannabis Use				0.38
Mean (SD)	2045 (3505.4)	1588 (2831.4)	1870.9 (3263.9)	
Median (IQR)	365.2 (2617.6)	182.6 (1964.2)	334.8 (2476.6)	
Age at First Cannabis Use				0.388
Mean (SD)	16.3 (5)	17 (6)	16.6 (5.4)	
Median (IQR)	15.5 (4.3)	16 (4.3)	16 (4)	
**METH Use Characteristics (*n* = 123)**	**PWoH (*n* = 73)**	**PWH (*n* = 50)**	**Total (*n* = 123)**	** *p* **
Past-month METH Use	27 (37.0%)	16 (32.0%)	43 (35.0%)	0.569
Lifetime DSM-IV METH Use Disorder	65 (89.0%)	42 (84.0%)	107 (87.0%)	0.414
Lifetime Days of METH Use				0.564
Mean (SD)	2457.3 (2447.9)	2206.3 (2234.9)	2355.3 (2357.5)	
Median (IQR)	1583 (2348)	1347 (3081.8)	1563 (2611.5)	
Lifetime METH Quantity (grams)				0.049
Mean (SD)	4092.5 (7649.4)	1859.6 (2605.8)	3184.8 (6202.6)	
Median (IQR)	1330.5 (4056.5)	1065.9 (1957.9)	1192.8 (2689)	
Days Since Last METH Use				0.678
Mean (SD)	199.6 (598.5)	248.2 (688.1)	219.4 (634.2)	
Median (IQR)	60.9 (168.6)	60.9 (214.3)	60.9 (168.6)	
Age at First METH Use				0.11
Mean (SD)	24.2 (11.2)	27.3 (8.9)	25.4 (10.4)	
Median (IQR)	21 (14)	25 (13)	23 (14.5)	

DSM-IV = American Psychiatric Association Diagnostic Statistical Manual IV, SD = standard deviation of the mean, IQR = interquartile ratio.

**Table 4 viruses-17-01143-t004:** Results from multivariable linear models examining main associations between lifetime MUD, past-month cannabis use, HIV status, and plasma markers, with retained covariates (top panel; Base Model) and results from models examining the main and conditional associations of lifetime MUD (MUD−/MUD+), past-month cannabis use (C−/C+), and HIV status (PWoH/PWH). Factor variables are expressed as the effect of going from one level to another (e.g., the effect of being male, relative to female; “Female -> Male”). All models were estimated using heteroscedasticity-robust standard errors. Model parameter estimates are standardized betas (B) with 95% confidence intervals (CI). Model R2 estimates are provided at each level of increasing model complexity. Conditional effects of lifetime MUD and past-month cannabis use are specified as simple comparisons with reference to the negative–negative group.

	CXCL10/IP-10		CCL2/MCP-1		ICAM-1		VCAM-1	
	B	95% CI	B	95% CI	B	95% CI	B	95% CI
* Base Model (Covariates-only) *								
Age	0.11 *	[0.02, 0.19]	0.31 ***	[0.19, 0.42]	0.11 *	[0.02, 0.19]	0.07	[−0.01, 0.15]
Sex: Female -> Male			0.40 *	[0.08, 0.72]	−0.17	[−0.36, 0.02]	−0.15	[−0.32, 0.03]
Ethnicity: White -> US Minority	0.17	[−0.00, 0.35]	0.19	[−0.09, 0.47]			−0.35 ***	[−0.51, −0.20]
Education Years					−0.12 **	[−0.20, −0.04]	−0.06	[−0.13, 0.01]
Past-month Tobacco Use					0.26 *	[0.05, 0.47]	−0.11	[−0.28, 0.06]
*R^2^*	0.08		0.16		0.13		0.19	
* Main Effects: Lifetime DSM-IV METH Use Disorder, Past-month Cannabis Use, and HIV Status *								
Lifetime DSM-IV METH Use Disorder	0.01	[−0.17, 0.18]	0.32 *	[0.05, 0.59]	0.10	[−0.12, 0.32]	0.25 **	[0.09, 0.41]
Past-month Cannabis Use	−0.33 **	[−0.54, −0.11]	−0.04	[−0.31, 0.22]	−0.09	[−0.31, 0.12]	−0.10	[−0.29, 0.08]
HIV Status: PWoH -> PWH	0.54 ***	[0.35, 0.73]	0.13	[−0.14, 0.40]	0.17	[−0.02, 0.37]	0.31 ***	[0.15, 0.47]
*R^2^*	0.19		0.17		0.13		0.22	
* Conditional Effects: Lifetime DSM-IV METH Use Disorder, Past-month Cannabis Use, and HIV Status *								
MUD−C− > MUD+C+			0.21	[−0.23, 0.65]	0.00	[−0.30, 0.30]	0.10	[−0.30, 0.50]
MUD−C− > MUD+C−			0.36 *	[0.08, 0.64]	0.26 **	[0.07, 0.45]	0.42 ***	[0.19, 0.66]
MUD−C− > MUD−C+			0.04	[−0.36, 0.44]	0.01	[−0.25, 0.28]	0.19	[−0.14, 0.51]
MUD−C− > MUD+C+: PWoH -> PWH							0.15	[−0.39, 0.69]
MUD−C−> MUD+C−: PWoH -> PWH							−0.44 **	[−0.78, −0.11]
MUD−C− > MUD−C+: PWoH -> PWH							−0.61 **	[−1.08, −0.14]
*R^2^*		-	0.18		0.17		0.26	

All continuous predictors are mean-centered and scaled by 1 standard deviation. The outcome variable is in its original units (within-plate standardized). Standard errors are heteroskedasticity robust. Displayed base model and main effects model estimates for VCAM-1, ICAM-1, and CCL2/MCP-1 were estimated from those levels of complexity and cannot be used with conditional effects estimates to calculate marginal effects. For estimates of marginal means, see Figures 2 and 3. *** *p* < .001; ** *p* < .01; * *p* < .05.

## Data Availability

Data are maintained in secure repositories at the HIV Neurobehavioral Research Program, University of California San Diego, and are available upon request via email at hnrpresource@ucsd.edu.

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
