# Peer review of "Cannabis Use Moderates Methamphetamine- and HIV-Related Inflammation: Evidence from Human Plasma Markers"

_viruses, 2025, doi:10.3390/v17081143_

Round 1
Reviewer 1 Report
Comments and Suggestions for Authors
This manuscript presents a novel and well-executed investigation into the moderating role of cannabis use on inflammation in individuals with HIV and methamphetamine use disorder. While the statistical modeling is robust, the discussion lacks sufficient mechanistic interpretation of cannabinoid action. The clinical implications of these biomarker shifts are underdeveloped. Including cannabinoid type/dose data and prospective outcomes could strengthen translational value. Here are my suggestions/ questions and answers can make the manuscript more potential and increase clarity:
1. How were cannabinoid types (e.g., THC vs. CBD) standardized or accounted for across users?
2. Were potential confounding effects of ART regimens on inflammation biomarkers statistically adjusted?
3. What is the clinical significance of reduced plasma VCAM-1 or ICAM-1 levels in terms of neurovascular outcomes?
4. How might tobacco use have interacted with cannabis effects, given its association with ICAM-1?
5. Was there correction for multiple comparisons in testing four biomarkers across interaction terms?
6. How generalizable are the findings given the limited diversity in the PWH subgroup (e.g., mostly male)?
7. Can causality be inferred from past-month cannabis use without pharmacokinetic confirmation or longitudinal data?
8. Were plasma biomarker levels cross-validated with CNS markers (e.g., CSF, imaging) for neuroinflammation?
9. How were polysubstance use and historical drug patterns (e.g., cocaine, opioids) controlled in the model?
10. Does the absence of cannabis dose-frequency detail limit interpretation of dose-response relationships?
Author Response
1. How were cannabinoid types (e.g., THC vs. CBD) standardized or accounted for across users?
Cannabis product/cannabinoid concentrations were not collected in the research projects that generated these data. This limitation is acknowledged in the manuscript's limitations section:
• “Regarding the operationalization and examination of substance use variables, we did not have information about the cannabinoid products our participants were using at the time of their reports. Cannabinoid composition (e.g., THC vs. CBD vs. other cannabinoid dominance) may influence the degree to which cannabis is associated with changes in plasma markers (e.g., THC extract products and whole plant matter produce unique constellations of biological responses). Further, conclusions herein may be specific to our analytic strategy of modeling recent cannabis use (i.e., in past month). Our sample contained few people with current cannabis use disorder, and it is plausible that a current use disorder may be associated with different plasma marker relationships than current non-use disorder levels of cannabis use. This concern is partially ameliorated by the fact that our past-month cannabis using subsample displayed greater lifetime amount and days of cannabis use and earlier ages at first cannabis use than those who did not use in the past month.”
To further clarify this issue, we have added the following statement to the limitations and future directions section, which addresses cannabis product variability and the need for better understanding of their potentially differential impacts:
• Importantly, there is currently wide variation in commonly used cannabis products, and it will be important for future research to elucidate the pharmacological differences between products and whether this influences subsequent biological cascades. Participants in this sample were all using cannabis recreationally prior to the availability of legal dispensaries, and data from such cannabis samples indicates that Δ9-THC was the predominant cannabinoid, with CBD levels being very low or undetectable [70,71].
Also, in response to a suggestion from reviewer 2, which was related to this comment, we have edited/added the following text to our discussion of VCAM-1 results:
• We did not collect data on the specific cannabis products used by participants and were unable to test hypotheses related to CBD exposure. Because our participants were recreational cannabis users in the southwestern US, who largely began using cannabis prior to the availability of legal dispensaries, Δ9-tetrahydrocannabinol (Δ9-THC) is likely the primary cannabinoid they consumed, followed distantly by CBD and Δ9-tetrahydrocannabivarin [70,71]. There is notable evidence of Δ9-THC displaying greater affinity for both CB1 and CB2 receptors than the other most common phytocannabinoids, implicating a wider array of potential cannabinoid pathways than previous research on CB2 receptor agonism, specifically.
2. Were potential confounding effects of ART regimens on inflammation biomarkers statistically adjusted?
This study’s models examined data from both people with and without HIV, and as such, ART-related variables applied only to a subset of the sample. Including ART regimen details in the main models would have required complex multilevel modelling, which was not feasible given the sample size and could have substantially reduced power and interpretability. To address concerns about ART as a potential confound, we conducted post hoc analyses examining whether length of time (in months) on current ART regimen was associated with plasma markers, using Pearson correlations. We found no significant relationships between length of current ARV exposure and plasma markers.
VCAM1 r = 0.11, t = 0.99, df = 84, p-value = .325
ICAM r = -0.12, t = -1.13, df = 84, p-value = .263
MCP1 r = 0.09, t = 0.83, df = 84, p-value = .411
IP10 r = -0.09, t = -0.84, df = 84, p-value = .401
Regarding the types of ARV regimens, of the 86 participants with HIV, 20 (23%) were on both PI and NRTI, 20 (23%) on NNRTI and NRTI, 37 (43%) on NRTI and II, 1 (1%) on PI and II, 1 (1%) on NNRTI and II, 1 (1%) on just NRTI, and 6 (7%) were on 3+ drugs. After excluding drug combinations with insufficient cell sizes for inferential testing, we conducted a post-hoc examination (1x3 ANOVA) of ARV regimen type and their association with examined plasma markers (below). We found no significant differences between regimen groups.
Mean (SD) | NNRTI/NRTI (N=20) | NRTI/II (N=37) | PI/NRTI (N=20) | Total (N=77) | p-value |
VCAM1 | 0.032 (0.551) | 0.178 (0.445) | 0.190 (0.591) | 0.143 (0.511) | 0.532 |
ICAM1 | 0.137 (0.631) | 0.004 (0.587) | 0.116 (0.667) | 0.067 (0.615) | 0.683 |
CCL2/MCP1 | -0.120 (0.690) | 0.138 (0.958) | 0.291 (1.454) | 0.111 (1.052) | 0.461 |
CXXCL10/IP10 | 0.229 (0.776) | 0.195 (0.529) | 0.323 (0.904) | 0.237 (0.700) | 0.806 |
3. What is the clinical significance of reduced plasma VCAM-1 or ICAM-1 levels in terms of neurovascular outcomes?
Reduced plasma levels of VCAM-1 and ICAM-1 suggest decreased endothelial activation and improved vascular integrity. Clinically, this might indicate a lower risk of neurovascular injury and downstream adverse effects, as elevated adhesion molecules have been associated with blood-brain barrier disruption and neuroinflammation, which are key contributors to neurocognitive impairment in both HIV and methamphetamine use. However, markers examined in this study are not currently considered as reliable risk indicators for future events, and we are not aware of data regarding whether reducing levels of these markers can influence clinical risk. In our discussion and conclusion, we opted not to equate higher/lower levels of examined plasma markers with real-world clinical outcome risk, as use of these plasma markers is largely confined to research, and we do not currently have prospective clinical outcome data for this sample, though it is an important next step that would enhance the clinical relevance of our findings.
4. How might tobacco use have interacted with cannabis effects, given its association with ICAM-1?
While tobacco use displayed a main association with ICAM-1, we explored this possibility at your suggestion and found no conditional association between past-month cannabis and past-month tobacco use (b=-0.19, se=0.20, t=-0.92, p=.36). The main reason for not focusing more heavily on tobacco use and its influence in these data was due to its collinearity with those who have lifetime METH use disorder, as a majority of past-month tobacco use is represented by those who use METH. Although we did not see evidence of an interaction in our sample, the frequent co-use of methamphetamine and tobacco, as well as co-use of these substances with cannabis, warrants future focused examination of their potential interactions.
5. Was there correction for multiple comparisons in testing four biomarkers across interaction terms?
We did not employ alpha corrections for the present analysis, as examining four outcome variables in separate models, and we viewed the analyses as parallel rather than family-wise tests requiring adjustment. For the interaction terms, we assessed significance at the omnibus level (i.e., whether the inclusion of interaction terms significantly improved model fit). For effect interpretation, we pre-specified only one set of pair-wise contrasts ameliorating the need to adjust for multiple contrast sets and preserving power. Nonetheless we acknowledge the importance of considering multiple testing and have interpreted the results with appropriate caution.
6. How generalizable are the findings given the limited diversity in the PWH subgroup (e.g., mostly male)?
Though overall, the demographic composition was fairly consistent with the broader population of PWH in the United States, as over 80% of new cases are accounted for by men, but you raise an important point about the potential of ruling out a sex-specific effect in PWH. The extent to which gender effects could be observed in and generalized to PWH was impacted by the small n of women with HIV, and we have added text to the limitations section, addressing this: “Our ability to detect and draw inferences from estimates related to biological sex among PWH, specifically, was limited by sample size (7% female).”
7. Can causality be inferred from past-month cannabis use without pharmacokinetic confirmation or longitudinal data?
We agree that causal inferences cannot be drawn from the current cross-sectional design and the absence of pharmacokinetic confirmation. To avoid overstating the findings, we have reviewed the manuscript for language suggesting causality and revised these instances to better reflect the observational nature of the data, most notably in the conclusions section: “Results from this study support prior findings that METH and HIV disease confer risk for negative outcomes via their influence over chronic inflammatory processes, and we provide novel evidence from human plasma samples that cannabis use is associated with reduce levels of immune and inflammatory molecules in the context of chronic METH use or HIV infection.” This change ensures the language is consistent with an associative, not causal, interpretation of the findings.
8. Were plasma biomarker levels cross-validated with CNS markers (e.g., CSF, imaging) for neuroinflammation?
We did not conduct parallel modeling of other data sources. CSF data were not available for many (46.5%) participants in this study, and this level of missingness made such analysis infeasible with the current study design. However, this is an important future direction and have noted it as such in the Limitations and Future Directions. We acknowledge that the lack of CNS-specific biomarkers limits the ability to directly infer central neuroinflammatory processes from peripheral plasma markers, and we have added this point to the limitations section of the manuscript.
9. How were polysubstance use and historical drug patterns (e.g., cocaine, opioids) controlled in the model?
We did not include cocaine and opioid use disorders as covariates in the models due to cocaine use disorder being almost entirely confined to people with MUD and opioid use disorder alone having no theoretical link to the examined plasma markers. The low overall rates of these use disorders in our sample are due in part to the study's exclusion criteria, which excluded participants with any past-year substance use disorder other than METH, cannabis, or DSM-IV alcohol abuse (alcohol dependence was excluded). Given the more proximal relevance of alcohol and tobacco to our sample, all models initially considered past-month tobacco use and lifetime alcohol use disorder as covariates.
10. Does the absence of cannabis dose-frequency detail limit interpretation of dose-response relationships?
The frequency and estimated amount of cannabis consumed over the participants’ lifetimes were collected, reported, and analyzed in models described in the manuscript Results section “Associations Between Lifetime Cannabis and METH Use Characteristics and Plasma Biomarkers”. We found no associations between these lifetime cannabis dose and frequency estimates and plasma markers, and our discussion does not contain inferences or interpretations of cannabis dose/response relationships. All interpretations were made regarding the recency or presence/absence of cannabis use. We acknowledge in the limitations section that the lack of cannabinoid type and concentration data may limit the specificity of our findings regarding cannabis exposure (more detail above in response to #1).
Reviewer 2 Report
Comments and Suggestions for Authors
Reviewer Comments:
As a basic molecular scientist, I am not a specialist in clinical brain research although I studied this clinical field in the past. I found this manuscript to be well-prepared and clearly written. It originates from a well-established and experienced clinical research group. The data are robust and convincingly support the authors’ hypothesis. The findings are consistent with previous literature while providing new, additional insights.
The Discussion section is extensive, especially in relation to the four inflammatory markers. However, the main discovery or central message of this manuscript could be made clearer. It would strengthen the impact of the work to distinguish more explicitly between what is already well-known in the clinical field and what represents a novel or most important finding in this study.
This reviewer has a few minor comments:
- Exclusion Criteria – HIV RNA >50 copies/mL:
The exclusion of participants with plasma HIV RNA levels >50 copies/mL is noted. It would be helpful if the authors could explain the rationale behind this criterion. One might assume that individuals with detectable viremia could have elevated inflammatory markers. Clarifying this point would help the reader understand the population selected for analysis. - Cannabinoid Type and Source:
Since cannabis includes a variety of compounds and can be derived either from the plant or synthesized in the lab, the inflammatory effects may vary significantly depending on the type of cannabinoids used. It would improve the manuscript to include a brief discussion—perhaps in the limitations section—about the predominant types of cannabinoids likely used by participants in both the PWH and PWoH groups. This clarification aligns with the sentence currently in the Discussion (VCAM-1 section): “characteristics (e.g., dose, type of cannabinoids used)” - Final Statement on Clinical Targeting of Cannabinoid Pathways:
The authors conclude that “Cannabinoid pathways may be worthwhile clinical targets for treating sequelae of chronic inflammatory conditions.” While this is an interesting and potentially impactful idea, it would benefit from additional explanation. Specifically, the authors could elaborate on the mechanistic rationale or clinical relevance behind this suggestion to better support this concluding remark.
Author Response
1. The Discussion section is extensive, especially in relation to the four inflammatory markers. However, the main discovery or central message of this manuscript could be made clearer. It would strengthen the impact of the work to distinguish more explicitly between what is already well-known in the clinical field and what represents a novel or most important finding in this study.
Thank you for this feedback. We have made extensive edits throughout the discussion to more clearly state where our hypotheses served to further evince prior findings (HIV+ and MUD having pro-inflammatory associations), as opposed to providing relatively more novel evidence (cannabis use moderating HIV+ and MUD associations; main association with CXCL10/IP-10).
2. Exclusion Criteria – HIV RNA >50 copies/mL: It would be helpful if the authors could explain the rationale behind this criterion. One might assume that individuals with detectable viremia could have elevated inflammatory markers. Clarifying this point would help the reader understand the population selected for analysis.
This is a good question – a majority of modern (post- ARV drug availability) human participants studies recruit samples of people who have been on ARV drugs for a meaningful period of time and who are currently virally-suppressed, evinced by values under the lower limit of RNA copy detection by routine screening (i.e., >50 copies/mL). Clinical research in many countries where ARV drugs are widely available has largely narrowed focus to medication adherence initiatives and HIV-related pathology that persists, despite treatment. Notable among these persistent conditions are those posited to result from exposure to chronic pro-inflammatory processes, such as HIV-related neurocognitive decline. Bloch et al. (2020), among others, provide a better account of this history:
“In the early years of the epidemic, management was focused on acute, potentially life-threatening AIDS-related complications. From initial monotherapy with first-generation antiretroviral therapy (ART), the development of combination highly active ART (HAART) allowed HIV control but ART toxicities, treatment adherence, and drug resistance emerged as major issues. Today, the availability of potent and tolerable ART has made viral suppression achievable in most people living with HIV (PLHIV), and clinicians are confronted with managing a chronic condition among an ageing population. The combination of diseases of ageing and the co-morbidities associated with HIV-infection, even when well controlled, results in a complex set of challenges for many older PLHIV. There is a growing appreciation that many non-AIDS-related co-morbidities are caused, at least in part, by persistent, low-grade immune activation, inflammation, and hypercoagulability, despite suppressive ART.”
3. Cannabinoid Type and Source:
Since cannabis includes a variety of compounds and can be derived either from the plant or synthesized in the lab, the inflammatory effects may vary significantly depending on the type of cannabinoids used. It would improve the manuscript to include a brief discussion—perhaps in the limitations section—about the predominant types of cannabinoids likely used by participants in both the PWH and PWoH groups. This clarification aligns with the sentence currently in the Discussion (VCAM-1 section): “characteristics (e.g., dose, type of cannabinoids used)”
At your suggestion, we have added the following paragraph to the limitations section:
“Importantly, there is currently wide variation in commonly used cannabis products, and it will be important for future research to elucidate the pharmacological differences between products and whether this influences subsequent biological cascades. Participants in this sample were all using cannabis recreationally prior to the availability of legal dispensaries, and data from such cannabis samples indicates that THC is the predominant cannabinoid, with CBD levels being very low or undetectable [72,73].”
Additionally, we have added text to the Discussion section you referenced that further details the relevance of our participants’ likely cannabinoid exposures to the properties they display at CB1 and CB2 receptors.
“We did not collect data on the specific cannabis products used by participants and were unable to test hypotheses related to CBD exposure. Because our participants were recreational cannabis users in the southwestern US, who largely began using cannabis prior to the availability of legal dispensaries, Δ9-tetrahydrocannabinol (Δ9-THC) is likely the primary cannabinoid they consumed, followed distantly by CBD and Δ9-tetrahydrocannabivarin [70,71]. There is notable evidence of Δ9-THC displaying greater affinity for both CB1 and CB2 receptors than the other most common phytocannabinoids, implicating a wider array of potential cannabinoid pathways than previous research on CB2 receptor agonism, specifically.”
4. Final Statement on Clinical Targeting of Cannabinoid Pathways:
The authors conclude that “Cannabinoid pathways may be worthwhile clinical targets for treating sequelae of chronic inflammatory conditions.” While this is an interesting and potentially impactful idea, it would benefit from additional explanation. Specifically, the authors could elaborate on the mechanistic rationale or clinical relevance behind this suggestion to better support this concluding remark.
Thank you for this suggestion. We have re-written the conclusions section to better situate our conclusion in the context of persistent HIV-related pathology in the era of effective and tolerable ARV drugs:
METH use disorder is highly prevalent in PWH, and both can have significant effects on immune function and pro-inflammatory processes that lead to significant central nervous system consequences, despite modern advances in anti-retroviral therapy effectiveness and tolerability. Results from this study support prior findings that METH and HIV disease confer risk for negative outcomes via their influence over chronic inflammatory processes, and we provide novel evidence from human plasma samples that cannabis use is associated with reduced levels of immune and inflammatory molecules in the context of chronic METH use or HIV infection (CCL2/MCP-1, VCAM-1, ICAM-1) and independent of METH use or HIV (CXCL10/IP-10). Associations between cannabis use and lower indices of inflammatory pathology from HIV and MUD point toward cannabinoid pathways as promising therapeutic targets that warrant further study.
Reviewer 3 Report
Comments and Suggestions for Authors
The abstract would benefit from the inclusion of numerical data, such as fold changes and percentage differences. I recommend replacing the keywords with alternative terms that were not used in the title or abstract to broaden the document's search visibility.
The Introduction is well-articulated, laying a solid foundation for understanding the primary focus of the study. Kudos to the authors for their efforts.
Please include the ethics approval number for the research project.
The Results and Discussion sections are scientifically robust and well-structured. The findings are presented clearly, facilitating logical comprehension. The mathematical modeling employed is appropriate and yields coherent results. Moreover, the authors acknowledge potential biases and limitations, which underscores their commitment to maintaining high standards in scientific discourse. I have no reservations about endorsing the acceptance of this manuscript; however, I suggest addressing the following areas:
1. Could you elaborate on whether standard laboratory facilities would be capable of conducting this evaluation? How do you envision this being applied in real-world settings?
2. It would be beneficial to assess the potential influence of sex on the findings to enhance the analysis.
3. To improve clarity, consider incorporating a summary image at the end of the Discussion section to encapsulate the key findings succinctly.
Author Response
1. The abstract would benefit from the inclusion of numerical data, such as fold changes and percentage differences.
We agree that including select numerical estimates could enhance interpretability, and we would appreciate guidance on which parameters to include and what might be most appropriate to condense or omit to accommodate these additions. At present, we opted to not include statistical parameters, as three of four model groups contained complex associations that are difficult to summarize accurately, and we needed to reduce our word count to meet the 200-word limit (currently at 199/200).
2. I recommend replacing the keywords with alternative terms that were not used in the title or abstract to broaden the document's search visibility.
Thank you for suggesting this – we have added “inflammatory signaling pathways”, “central nervous system stimulant”, and “polysubstance use” to the keyword list.
3. Please include the ethics approval number for the research project.
Apologies for the omission – we provided the institutional review board approval number and consent forms to the editor following the initial submission.
4. Could you elaborate on whether standard laboratory facilities would be capable of conducting this evaluation? How do you envision this being applied in real-world settings?
At present, assays for these markers (e.g., ICAM-1, VCAM-1) are primarily used in research settings due to cost limited clinical validation for risk event estimation for real world clinical outcomes. Our purpose in collecting these data was to examine possible evidence of mechanistic connections between types of substance use, HIV disease, and indices of inflammation. Practically, we envision these methods contributing to a better understanding of the biological pathways affected by methamphetamine and HIV disease, in the context of a growing literature on chronic inflammatory conditions, which may ultimately guide the development of targeted interventions. Ideally, these assays will be used in future longitudinal studies of people with chronic inflammatory conditions and who do or do not use cannabis products. Should those studies suggest the same link between cannabis use and lower levels of circulating inflammatory markers, future randomized control trials may follow that better isolate the possible mechanisms behind the observed effects. It is plausible that, over the course of researching these markers, a connection between these marker profiles and certain pathologies may arise, in which case they may be used in real-world clinical settings.
5. It would be beneficial to assess the potential influence of sex on the findings to enhance the analysis.
We initially examined sex at birth as a covariate in all models, and all presented model results can be interpreted as having statistically controlled for the influence of sex. We retained sex at birth in all models but CXCL10/IP-10, as it displayed no association with CXCL10/IP-10 and did not improve model fit. Otherwise, men displayed significantly higher MCP-1 levels and relatively (but significantly) lower levels of ICAM-1 and VCAM-1. Only the association with CCL2/MCP-1 met full criteria for statistical inference, and this was located in the Modeled associations… section of the manuscript – line 278. The lack of consistency in findings related to sex at birth discouraged us from attempting to synthesize them in our discussion, and another reviewer also raised the valid point that our findings related to sex are limited in their generalizability to women with HIV, as our sample contained relatively few (7%). We have added text to the limitations section explaining this consideration, and thank you for raising this point, as well.
6. To improve clarity, consider incorporating a summary image at the end of the Discussion section to encapsulate the key findings succinctly.
We have thoroughly edited the discussion and conclusions sections at suggestions from the other reviewers, but we were not able to generate a novel figure that appropriately supplements the discussion and conclusion without redundancies to other manuscript figures. We do not intend for these data to directly evince distinct biological mechanisms, which is why we have opted to not include artistic renderings of the immune and inflammatory pathway models.
Reviewer 4 Report
Comments and Suggestions for Authors
This manuscript addresses the disproportionately high prevalence of methamphetamine (METH) use among people living with HIV (PLWH), emphasizing its role in exacerbating cardiovascular and neurocognitive pathology through the induction of a chronic inflammatory state. The primary objective of this study was to examine potential interactions between lifetime methamphetamine use disorder (MUD), recent cannabis use, and HIV status on four plasma biomarkers indicative of immune and inflammatory function. The authors propose that cannabis use may attenuate immune and inflammatory dysregulation associated with METH exposure or HIV infection, suggesting that cannabinoid pathways could serve as valuable clinical targets for mitigating inflammatory disease sequelae.
Overall, the manuscript is well-written and supported by appropriate evidence. The study was well conducted, and the data were well analyzed and presented Nevertheless, I have several recommendations that could enhance clarity and rigor:
- Line 25 – Define all abbreviations upon first use to facilitate better flow and comprehension for readers.
- Line 45 – Specify the exact quantity that constitutes a “substantial portion” to avoid ambiguity.
- Line 158: “(Robins et al., 1988)” must be cited as a number and added to the list of references.
- Ensure that all abbreviations are defined only once at their initial appearance and consistently used throughout the text.
- In the introduction, while discussing plasma markers of immune activation, the authors should explicitly mention CXCL10/IP-10, which is a key biomarker in this study and is currently omitted from this section.
- For clarity, include the full meaning of abbreviations in the footnotes of all tables.
- The data presentation within tables becomes confusing in certain sections. I suggest inserting dividing lines to clearly separate different categories or variables.
- Add appropriate legends and symbol explanations to all figures. Additionally, adjust figure sizes to fit seamlessly within the text layout.
- Standardize the reference formatting to ensure consistency throughout the manuscript.
- The list of abbreviations, inserted before references, is incomplete.
Author Response
1. Line 25 – Define all abbreviations upon first use to facilitate better flow and comprehension for readers.
We did not provide the full names of examined plasma marker molecules in the abstract due to the word count restrictions. As submitted, the abstract is 199/200 words, but we will expand the names with approval from the editor.
2. Line 45 – Specify the exact quantity that constitutes a “substantial portion” to avoid ambiguity.
We have substituted a National Survey on Drug Use and Health cross-tabulation estimate for the previous citation, which did not provide precise estimates. The exact percentage is variable in community and clinical samples but typically range from 50-100%.
3. Line 158: “(Robins et al., 1988)” must be cited as a number and added to the list of references.
Thank you for pointing this out – we have corrected this reference.
4. Ensure that all abbreviations are defined only once at their initial appearance and consistently used throughout the text. For clarity, include the full meaning of abbreviations in the footnotes of all tables.
In addition to ensuring that the back matter abbreviation list is complete, we have reviewed the manuscript and removed redundant/ operationalizations in-text. For tables, we have added operationalizations to the table descriptions and footnotes.
5. In the introduction, while discussing plasma markers of immune activation, the authors should explicitly mention CXCL10/IP-10, which is a key biomarker in this study and is currently omitted from this section.
We apologize for the inconsistencies in the introduction - CXCL10/IP-10 was originally mentioned twice in the introduction, using the shorter naming convention “IP-10”, which we failed to correct before the initial submission. We have now edited these mentions to state “CXCL/IP-10” to be consistent with the rest of the manuscript.
6. The data presentation within tables becomes confusing in certain sections. I suggest inserting dividing lines to clearly separate different categories or variables. Add appropriate legends and symbol explanations to all figures. Additionally, adjust figure sizes to fit seamlessly within the text layout.
Thank you for this feedback - figure captions have been expanded for clarity and figures have been simplified in places to allow for small image sizes. In addition, we have adjusted the document formatting to ensure that all tables and figures have been assigned the correct document properties and are displayed optimally within PDF constraints. We will work with the copy-editing team to make sure our visual media are displayed effectively and that formatting requirements are met.
Round 2
Reviewer 1 Report
Comments and Suggestions for Authors
The manuscript may be accepted in it's current form.